# Human Pluripotent Stem Cell-Derived Alveolar Organoid with Macrophages

**DOI:** 10.3390/ijms23169211

**Published:** 2022-08-16

**Authors:** Ha-Rim Seo, Hyeong-Jun Han, Youngsun Lee, Young-Woock Noh, Seung-Ju Cho, Jung-Hyun Kim

**Affiliations:** 1Division of Drug Evaluation, New Drug Development Center, Osong Medical Innovation Foundation, Cheongju-si 28160, Korea; 2Division of Intractable Diseases, Department of Chronic Diseases Convergence Research, Korea National Institute of Health, Cheongju-si 28159, Korea; 3Korea National Stem Cell Bank, Cheongju-si 28159, Korea

**Keywords:** human pluripotent stem cells, lung organoid, macrophage, differentiation, inflammation

## Abstract

Alveolar organoids (AOs), derived from human pluripotent stem cells (hPSCs) exhibit lung-specific functions. Therefore, the application of AOs in pulmonary disease modeling is a promising tool for understanding disease pathogenesis. However, the lack of immune cells in organoids limits the use of human AOs as models of inflammatory diseases. In this study, we generated AOs containing a functional macrophage derived from hPSCs based on human fetal lung development using biomimetic strategies. We optimized culture conditions to maintain the iMACs (induced hPSC-derived macrophages) AOs for up to 14 days. In lipopolysaccharide (LPS)-induced inflammatory conditions, IL-1β, MCP-1 and TNF-α levels were significantly increased in iMAC-AOs, which were not detected in AOs. In addition, chemotactic factor IL-8, which is produced by mononuclear phagocytic cells, was induced by LPS treatment in iMACs-AOs. iMACs-AOs can be used to understand pulmonary infectious diseases and is a useful tool in identifying the mechanism of action of therapeutic drugs in humans. Our study highlights the importance of immune cell presentation in AOs for modeling inflammatory pulmonary diseases.

## 1. Introduction

Human Pluripotent stem cells (hPSCs) can be differentiated using techniques that replicate in vivo organogenesis to create highly organized three-dimensional (3D) tissues that resemble the structure and function of organs [1]. During development, epithelial and mesenchymal progenitor cells containing organ buds first appear [2]. Through numerous branching and extensions, these embryonic organ buds develop into mature organs [3]. In 3D culture systems, mimicking epithelial-mesenchymal interactions essential for organogenesis during embryonic development can be recapitulated to direct the production of comparable tissue-like structures in vitro [4].

Human lung organoids derived from hPSCs were first reported by Dy e et al. [5]. The organoid contained several alveolar cells, such as the proximal airway epithelium, distal alveolar epithelium, mesenchymal lineages of cells, and recapitulated naïve human lung cells. Since then, several researchers have reported technologies for the generation of alveolar organoid (AOs) mixed with epithelial cell lines, such as alveolar epithelial cells (AEC)1, AEC2, and basal alveolar stem cells [6,7]. Such organoids have been used to model pulmonary diseases including pulmonary fibrosis and the COVID-19 infection [8,9]. To date, AOs derived from hPSCs have been shown to recapitulate lung functions.

However, most lung organoids do not contain immune cells, which limits the recapitulation of the immune response in lung organoids [9,10,11]. Recently, there has been a report of macrophages (iMACs) containing AOs (iMACs-AOs), but the location of these iMACs was outside the alveolar surface, which is in the opposite direction to that of human lung tissue [12].

Through the intake of gases, particles, suspended poisons and allergens, and infectious agents, the lungs are exposed to various hazardous external surroundings [13]. As a result, our immune system is developed to defend against such insults, without harming the lung tissues [14]. To process and eliminate potentially dangerous inhaled compounds and protect the pulmonary microenvironment, the respiratory tract is lined with mononuclear phagocytic cells (MPCs). These cells have a high level of phagocytic activity, release many mediators, and mobilize the proper immune response to counteract damaging assaults from external agents [15]. These MPCs are mostly dispersed in alveolar tissues, lung tissues, and airways, with over 90% of them being macrophages. The macrophages in the lung are either acquired during development or are recruited when the lungs sustain an injury [11].

Here, we present a method we developed to generate a lung organoid containing iMACs inside the alveolar sac from hPSCs to mimic the human lung, while considering its microenvironment. We established a protocol for differentiating 3D lung organoids and iMACs from hPSCs. To generate iMACs-AOs, we developed a technique to inject macrophages into 3D lung organoids. In conclusion, iMACs coexisted in AOs for a long period of up to 14 days and released cytokines under inflammatory conditions. iMACs-AO is a multi-cellular organoid that better recapitulates the human fetal lung, which will provide an advanced understanding of the pathogenesis of human lung diseases.

## 2. Results

### 2.1. Construction of 3D Lung Organoid Differentiation

We established a step-by-step differentiation protocol to generate hPSCs-derived alveolospheres (Figure 1A). First, the hPSCs were differentiated into a definitive endodermal lineage. Following efficient anteriorization, the cells were further differentiated into lung progenitor cells in CBRa medium (Figure 1B). Then, the NKX2.1-positive lung progenitors, CD47hiCD26lo cells (approximately 10% of total cells), were isolated using fluorescence-activated cell sorting (FACS) (Figure 1C), made to exist as single cells in a 3D environment, and induced to differentiate into lung organoids (Video S1). As a result, differentiation continued for 30 days, and lung organoids were successfully formed (Figure 1B). The diameter of the lung organoids gradually increased over time after cell isolation, and the diameter of the lung organoids formed on day 30 of differentiation was approximately 500–800 nm (Figure 1D). Notably, both a human embryonic stem cell line (hESC, H9) and a human-induced pluripotent stem cell line (hiPSC, CMC009)-derived alveolar organoid showed similar sizes (Figure 1D).

### 2.2. Characterization of Differentiated Lung Organoids

During differentiation, the expression of SRY-box transcription factor 17 (SOX17), a definitive endoderm marker, was the highest on day 4 of differentiation and then gradually decreased in a time-dependent manner. The expression of alveolar epithelial cell markers pulmonary surfactant-associated protein B (SFPTB) and pulmonary surfactant-associated protein C (SFPTC) increased during differentiation (Figure 2A). Immunofluorescence staining confirmed that lung organoids expressed the lung-specific markers SFPTC and NK2 homeobox 1 (NKX2.1), and forkhead box protein J1 (FoxJ1; an AEC type1 marker) and podoplanin (PDPN; as an AEC type2 marker) were found (Figure 2B) in some structures of the lung organoids.

### 2.3. Differentiation of iMACs from the hPSCs

hESCs were differentiated into iMACs according to a previously reported method [16] (Figure 3A). The morphology of iMACs was comparable to that of typical blood monocyte-derived macrophages (Figure 3B). FACS analysis revealed that 97.9% of iMACs were double-positive for the macrophage-specific markers CD14 and CD45, and 87.9% of iMACs were double-positive for CD86 and CD45 (Figure 3C) [17]. To evaluate whether iMACs are functional, we performed a phagocytosis assay. When we incubated iMACs with opsonized beads, which expressed GFP, 95% cells were GFP+. These data indicated that iMACs are phagocytic cells (Figure 3D).

### 2.4. Introduction of iMACs into the Lung Organoid

After the differentiation of lung organoids and iMACs from hPSCs, we introduced iMACs into lung organoids. iMACs were injected directly into the lung organoids using a syringe (Video S2). The injected iMACs remained in the lung organoid for 14 days (Supplemental Appendix A). The structure of the lung organoid became complicated after iMACs insertion and budding from the edge was observed (Figure 4A). After 14 days, the lung organoids were sectioned for immunofluorescence and the interior of the organoids was analyzed. CD14/CM-DiI or CD45/CM-DiI double-positive cells were present inside the lung organoid (CD14 and CD45: macrophage markers, CM-DiI positive: pre-stained macrophages, SFTPC: lung cell marker) (Figure 4B). In addition, 7 days after injecting iMACs into the lung organoid, the expressions of characteristic markers iMACs including osteopontin (OPN) [18], monocyte chemoattractant protein-1 (MCP-1) [19], macrophage inflammatory protein 1-β (MIP-1β) [20], and tissue inhibitor of metalloproteinases 2 (TIMP-2) [21] were increased in the growth medium (Figure 4C). Therefore, iMACs were successfully injected into lung organoids and maintained their function for at least 14 days.

### 2.5. Morphological and Molecular Changes of Lung Organoids with iMACs

To understand the response of lung organoids containing iMACs to inflammation, after successfully preparing lung organoids containing iMACs, lipopolysaccharide (LPS) was administered to induce an inflammatory reaction. iMACs were present in lung organoids during the induction of the inflammatory response (Figure 5A). The inflammatory response was induced in lung organoids seven days after the injection of macrophages. The expression of interleukin 1 beta (IL-1β) and tumor necrosis factor alpha (TNF-α) genes increased during the inflammatory reactions (Figure 5B) in the iMAC containing AO compared to the AO without iMAC, suggesting the importance of immune cells in pulmonary inflammation. When the inflammatory response was induced in iMACs, in contrast to the increased expression of TNF-α in the iMACs culture medium, there was no increased expression of TNF- α in the lung organoid medium (Figure 5C). As shown in Figure 5D, the expression of TNF-α protein did not increase in the lung organoid-embedded iMACs when placed in lung organoid culture medium, even though an inflammatory reaction was induced by LPS. The secretion of MCP-1, OPN, iInterleukin 8 (IL-8), and MIP-1β proteins was increased in lung organoid-embedded iMACs compared to that in lung organoids not embedded with iMACs. Additionally, IL-8 and MIP-1β levels were further increased when an inflammatory reaction was induced by LPS (Figure 5D). Therefore, the increased secretion of MCP-1, OPN, IL-8, and MIP-1β was due to embedding of iMACs into lung organoids, and the increase in IL-8 and MIP-1β was due to an inflammatory response.

## 3. Discussion

We inserted iMACs into lung organoids and observed their effects on inflammatory conditions. The primary findings of our study were as follows: (1) the lung organoid differentiation method was successfully established; (2) iMACs were directly injected into the lung organoid to confirm morphological changes; (3) iMACs that entered the lung organoid could coexist for 14 days; and (4) when the inflammatory response was induced, the secretion of IL-8 and MIP-1β in lung organoid-embedded iMACs increased.

Recently, the development of 3D in vitro differentiation of human AOs from hPSCs has provided more precise modeling of human lung diseases and drug screening, compared to those from 2D culture systems. However, the lack of immune cells is considered to be a limitation of this system. This model system can better recapitulate human airway diseases in the context of cellular diversity than AOs without immune cells.

A live cell-detecting dye-staining assay confirmed that iMACs survived for 14 days once inserted into the lung organoids (Supplemental Appendix A, Figure 4A). iMACs in lung organoids have been found to exist in complex structures. This result is significant as it was discovered for the first time that iMACs remain in lung organoids for a long time.

Specifically, after establishing the microenvironment of the lungs, we induced an inflammatory reaction. The expression of IL-8 and MIP-1β was increased compared to that in organoids without iMACs. IL-8 and MIP-1β are pro-inflammatory chemokines that rapidly enhance the immune response [22,23,24]. In particular, the expression of IL-8 and MIP-1β increases when an immune response occurs in iMACs and epithelial cells [24,25]. Therefore, our results are consistent with those of previous studies.

As shown in Figure 5C, in contrast to the increased expression of TNF-α protein when the iMACs were in the macrophage culture medium, the expression of TNF-α protein did not increase in the lung organoid medium, even after inducing an inflammatory response in iMACs by LPS. Lung organoid medium was composed of dexamethasone, cyclic adenosine monophosphate (cAMP), 3-isobutyl-1-methylxanthine (IBMX), and ITS premix (including selenium). Glucocorticoids (dexamethasone) suppress inflammation through multiple mechanisms, resulting in an immune response to resolve the inflammatory reactions [26,27]. IBMX, a non-specific cyclic nucleotide phosphodiesterase inhibitor, elevates intracellular cAMP and cyclic guanosine monophosphate and inhibits TNF-α synthesis, thereby reducing inflammation [28,29,30,31]. Intracellular cAMP predominantly suppresses innate immune functions of monocytes, macrophages, and neutrophils [32,33]. In addition, selenium is an essential micronutrient that suppresses redox-sensitive transcription factor NF-κB-dependent pro-inflammatory gene expression and induces anti-inflammatory effects in iMACs [34]. Although the inflammatory response was induced by LPS, the expression of TNF-α in lung organoid-embedded macrophages was suppressed, and the inflammatory response was weakly induced by supplementation with lung organoid culture medium. We are currently investigating this issue in disease modeling using lung organoid-embedded iMACs.

Along with immune cells, the vasculature is another important microenvironment of the respiratory system. Although our study has constructed an advanced microenvironment by developing a method to generate a macrophage-containing alveolar structure, to better recapitulate the airway, a vascularized AO with immune cells is needed.

Due to difficulties in accessing human lung tissues, there have been limitations in studying the pathogenesis of lung diseases. hPSCs- derived alveolar organoids containing immune cells provide an alternative cell source to study the molecular pathogenesis, genetic screening and disease pathways of diverse lung diseases. Idiopathic pulmonary fibrosis (IPF) is the most common and lethal form of lung disease. Previously, Chen et al. recapitulated the IPF phenotype using lung organoids [35]. In addition, lung organoids derived from hPSCs have been used to study virus infections such as RSV, HPIV3, and COVID-19 [35,36]. As interest in lung diseases caused by COVID-19 has increased, various diagnostic platforms and respiratory infectious disease modeling systems are being developed for the treatment of lung diseases [37]. Although these studies demonstrate that lung organoids can be adopted to model diverse pulmonary diseases, a limitation of the organoids used in previous studies was the lack of immune cells.

In this study, we constructed a lung microenvironment similar to that of an existing lung simulation. Therefore, the use of this platform will allow a deeper understanding of the pathogenesis of pulmonary diseases, especially those mediated by innate immune responses. In addition, mass production is possible with high throughput, which will help in the development of new drugs to treat lung diseases.

## 4. Materials and Methods

### 4.1. hPSCs Culture and Maintenance

H9 hESCs were purchased from WiCell (Madison, WI, USA). CMC009 hiPSCs were obtained from the Korean National Stem Cell Bank (http://kscr.nih.go.kr). Both cell lines were maintained in TeSR-E8 medium (05990, STEMCELL Technologies, Vancouver, Canada) on tissue culture plates coated with growth factor-reduced Matrigel (354230, BD).

### 4.2. hPSCs Differentiation into Alveolar Organoids

To acquire definitive endoderm cells, both of the hESC (H9) and hiPSC (CMC009) were seeded on a Matrigel-coated culture plate at a density of 2.1 × 10^5^ cells/cm^2^ and differentiated using a definitive endoderm differentiation kit (05110, STEMCELL Technologies). After five days, the cells were detached using Accutase (A6964, Sigma-Aldrich, St. Louis, MO, USA) and split from 1 to 4. Cells were treated with Y-27632 (10 μM, 1254, Tocris, Bristol, UK) for the first 24 h to allow the cells to attach to the plate. For anteriorization, cells were treated with SB431542 (10 μM, 1614, Tocris) and dorsomorphin (2 μM, P5499, Sigma-Aldrich) for 5–7 days of differentiation. Then, they were treated in the “CBRα medium” for 7–15 days of differentiation. CBRα medium was composed of CHIR99021 (3 μM, 4423, Tocris), BMP4 (10 ng/mL, 120-05, Peprotech, London, UK), and retinoic acid (100 nM, R2625, Sigma-Aldrich). On differentiation day 15, cells were detached from Accutase and sorted into CD47hiCD26lo cells. The isolated cells were mixed with Matrigel at a ratio of 1:2 and 60 μL was added to each well to form a three-dimensional environment. The drop was incubated for 30 min, the “lung organoid medium” was added, and Y-27632 was added for the first 24 h. AOs medium was composed of CHIR99021, SB431542, dexamethasone (50 nM, D4902, Sigma-Aldrich), cAMP (100 μM, B7880, Sigma-Aldrich), IBMX (100 μM, I5879, Sigma-Aldrich), keratinocyte growth factor (100 ng/mL, 251-KG-010, R&D Systems, Minneapolis, MN, USA), CaCl2 (800 μM, C7902, Sigma-Aldrich), and ITS Premix (354351, Corning, Corning, NY, USA). Fifteen days post-plating, organoids were detached and harvested for characterization assays [38].

### 4.3. hESCs Differentiation into Macrophages (iMAC)

Differentiation of hESCs into iMAC was performed as previously described [16]. Briefly, 5–10 colonies were seeded into a 35 mm dish on differentiation day −1. After a day, the cells were exposed to APEL2 medium containing BMP4 for 4 days for mesoderm differentiation. On day 4, mesodermal cells were further differentiated into a hematopoietic stem and progenitor cells using vascular endothelial growth factor (VEGF) and stem cell factor (SCF). After 2 days, the medium was replaced with a hematopoietic differentiation medium supplemented with insulin-transferrin-selenium, SCF, thyroid Peroxidase (TPO), Interleukin 3 (IL-3), Interleukin 6 (IL-6), and fms-related tyrosine kinase 3 ligand (Flt-3L). On differentiation days 15–30, floating cells were transferred into fetal bovine serum (FBS)-coated dishes, and iMACs differentiation was induced using MCSF and RPMI 1640 containing 10% FBS-contained RPMI 1640. At 15–50 days post-plating, the cells were detached and harvested for the assay. The cells were cultured in a humidified incubator with 5% CO_2_.

### 4.4. Phagocytosis Assay

The phagocytosis assay using iMACs was performed as previously described [16]. Briefly, the carboxylate-modified fluorescein isothiocyanate-conjugated latex beads with a mean diameter of 1.0 μm (Sigma-Aldrich) were cultivated with floating iMACs at a ratio of 1:0.2 (cell: beads) to 1:200 for 1.5 h. After washing to three times, phagocytic cells were examined using fluorescence-activated cell sorting (LSRFortessa; GFP) and a microscopy (Imager A2; Carl Zeiss AG, Oberkochen, Germany).

### 4.5. Generation of iMACs Containing AOs

On day 30 of lung organoid differentiation, hPSC-derived macrophages were stained with CellTracker^TM^ CM-DiI (C7000, Invitrogen, Waltham, MA, USA). Live staining was performed according to the manufacturer’s instructions. The macrophages were then injected into the lung organoid at a concentration of 2.0 × 10^7^ cells/mL using a 0.3 mL insulin syringe. Fluorescence was continuously observed after the injection on day 45 for organoid differentiation. During the macrophage injection stage, the cell-cultured medium used a “lung organoid medium”.

### 4.6. Quantitative Real-Time Polymerase Chain Reaction (qRT-PCR)

All samples were extracted with TRIzol reagent (15596018, Invitrogen) and eluted with messenger ribonucleic acid (mRNA) according to the manufacturer’s recommendations. The mRNA concentration was measured using Nanodrop (ND-2000, Thermo Scientific, Waltham, MA, USA), and 1 μg of mRNA was used for complementary deoxyribonucleic acid (cDNA) synthesis along with the PrimeScript™ RT reagent kit (RR037A, Takara, Kusatsu, Japan) at 37 °C for 15 min. qRT-PCR was performed using the Power SYBR™ Green PCR Master Mix (4367659, Applied Biosystems, Waltham, MA, USA). Gene expression was detected using the Applied Biosystems^®^ 7500 Real-Time PCR System (Applied Biosystems). Relative gene expression levels were quantified using the ΔΔCt method and were normalized to that of GAPDH. The qPCR primers, which included intron-spanning primers, were designed using the Probe Finder software (Version 2.50, https://www.roche-applied-science.com, Roche, Basel, Switzerland).

### 4.7. Immunofluorescence Staining

Organoids were washed with Dulbecco’s phosphate-buffered saline (DPBS) and fixed with 1% paraformaldehyde (P6148, Sigma) in DPBS at room temperature for 10 min. Fixed organoids were embedded in Tissue-Tek OCT compound (4583, Sakura) for cryo-sectioning. Frozen tissues were cut using a cryostat and washed with DPBS. The sections were permeabilized with 0.5% Triton X-100 (X-100, Sigma) in DPBS for 5 min and blocked with blocking reagent (11096176001, Roche) in 0.1% Tween 20 in DPBS (PBST) at room temperature for 1 h. The sections were incubated overnight in a cold humid chamber with the following antibodies: anti-CD14 (ab182032, Abcam, Cambridge, UK), anti-CD45 (ab10558, Abcam), anti-PDPN (ab10288, Abcam), anti-FoxJ1 (ab235445, Abcam), mature SFTPC (WRAB-76694, Seven Hills Bioreagents, Cincinnati, OH, USA), and anti-NKX2.1 (ab72876, Abcam). After incubation, the sections were washed thrice with PBST and incubated for 1 h with the following antibodies: Alexa Fluor^®^ 488 goat anti-rabbit IgG (A11008, Molecular Probes, Eugene, OH, USA), Alexa Fluor^®^ 647 goat anti-rabbit IgG (A21245, Molecular Probes), and Alexa Fluor^®^ 488 goat anti-mouse IgG (A11001, Molecular Probes). The nuclei were stained with Hoechst 33,342 (H3580; Invitrogen). The stained sections were washed thrice and mounted with ProLong^TM^ Gold antifade reagent (P36934, Invitrogen). Immunofluorescence images were acquired using a Zeiss confocal fluorescence microscope (LSM710, Carl Zeiss, Oberkochen, Germany).

### 4.8. Flow Cytometry

Cells were harvested using TrypLE™ Express Enzyme (12604013, Gibco, Waltham, MA, USA) and fixed using a Cytofix/Cytoperm™ Fixation/Permeabilization Kit (554714, BD). The cells were incubated at 4 °C for 45 min with anti-CD47 (323110, BioLegend, San Diego, CA, USA), anti-CD26 (302705, BioLegend), anti-CD45 (557883, BioLegend), anti-CD14 (565283, BioLegend), and CD86 (564544, BioLegend) antibodies. Stained cells were analyzed using a flow cytometer (FACS Canto or FACS Aria III cell sorter, Becton Dickinson). Data acquisition and processing were performed using FACS Diva software (Becton Dickinson).

### 4.9. Cytokine Array

The experiments were performed according to RayBio^®^ Human Cytokine Array C5 (AAH-CYT-5, Peachtree Corners, RayBiotech, GA, USA) guidelines. Briefly, the array membranes were stabilized at room temperature and incubated with a blocking buffer for 30 min. After blocking, the membranes were incubated with the organoid supernatant at room temperature. The membrane was washed five times and incubated with the antibody mixture for 2 h. After the addition of streptavidin antibody for 2 h, the membranes were detected using detection buffers C and D. The data were normalized using blots of positive and negative controls. Chemiluminescence was visualized using the ChemiDoc imaging system (Bio-Rad, Hercules, CA, USA).

### 4.10. Statistics

The statistical significance of the experimental results is presented as the mean ± standard deviation. The significance of the difference between the means was analyzed using one-way analysis of variance (Student–Newman–Keuls test). Statistical significance was set at ** p* < 0.05. All statistical analyses were performed using SigmaStat 3.5 (Systat Software Inc.,San Jose, CA, USA).

## 5. Conclusions

hPSCs-derived alveolar organoids with macrophages:A method for lung organoid differentiation has been successfully established;To verify the morphological changes, iMACs were injected directly into the lung organoids;For 14 days, iMACs coexisted in the lung organoids;IL-8 and MIP-1 release in lung organoid-embedded iMACs increased when the inflammatory response was triggered.

## Figures and Tables

**Figure 1 ijms-23-09211-f001:**
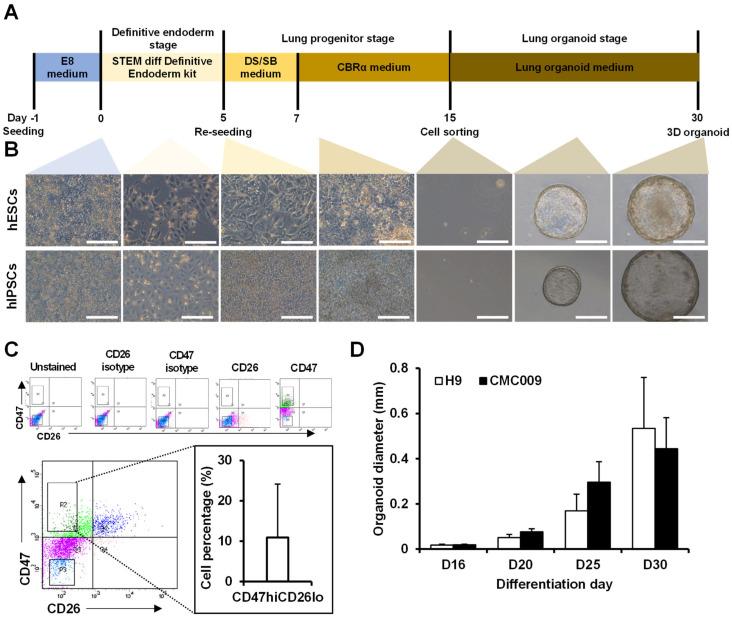
Generation of 3D lung organoids from hPSCs. (**A**) Schematic diagram of lung progenitor cell differentiation and 3D lung organoid generation from hPSCs. (**B**) The representative phase contrasts images of lung progenitor cells and lung organoids from hESCs and hIPSCs at differentiation days 0 to 30. Scale bar = 400 μm. (**C**) The cell percentage of CD47hi CD26lo population. *n* = 5. (**D**) Measurement of organoid diameter at differentiation day 16 to 30. The error bar depicts standard deviation. *n* = 20.

**Figure 2 ijms-23-09211-f002:**
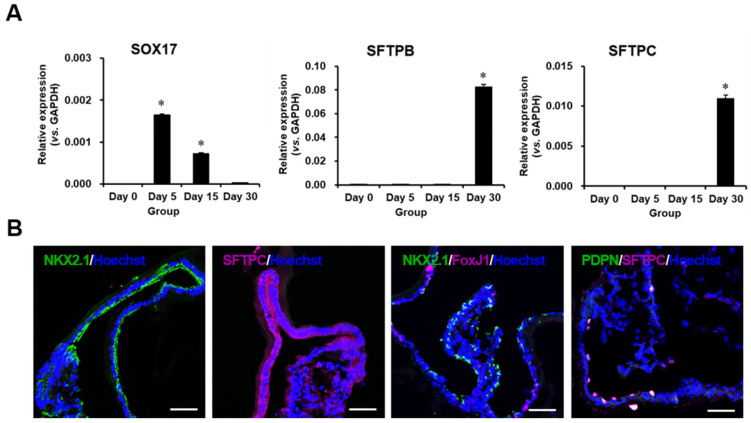
Phenotypic characterization of lung organoid. (**A**) Representative qRT-PCR analysis for SOX17, SFTPB, and SFTPC expression of differentiated cells at differentiation days 0 to 30. *n = 3.* * *p* < 0.05 versus day 0. (**B**) Immunofluorescence images of lung organoid at day 30. Scale bar = 50 μm.

**Figure 3 ijms-23-09211-f003:**
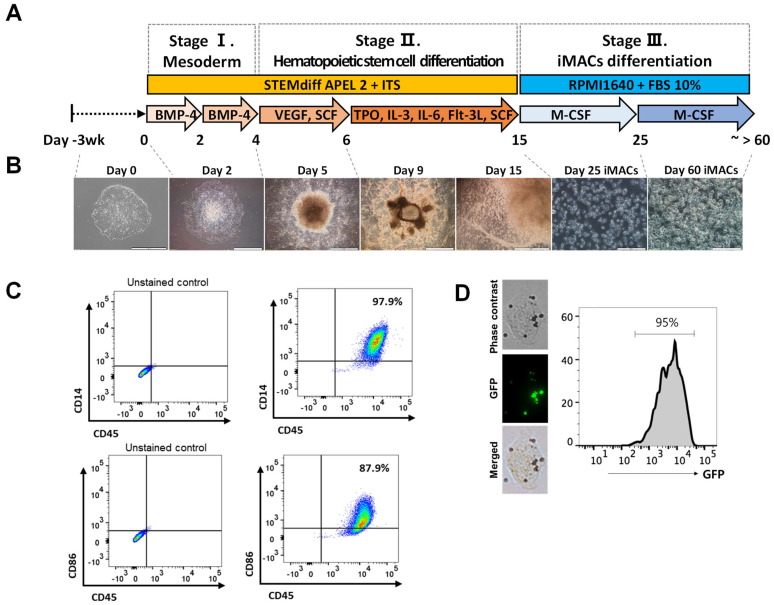
Step-wise macrophage differentiation and characterization of hPSCs-derived macrophages. (**A**) Schematic protocol of the step-wise macrophage differentiation from the hPSCs. (**B**) Morphological changes of the differentiated cells in each step of differentiation. Step 1 ~ 2 scale bar: 1 mm, Step 3 scale bar: 200 μm. (**C**) Flow cytometric analysis of marker gene expression (CD14, CD45, and CD86). (**D**) The single cell image of uptake of opsonized latex-beads by phagocytic cells. % uptake of beads by cells.

**Figure 4 ijms-23-09211-f004:**
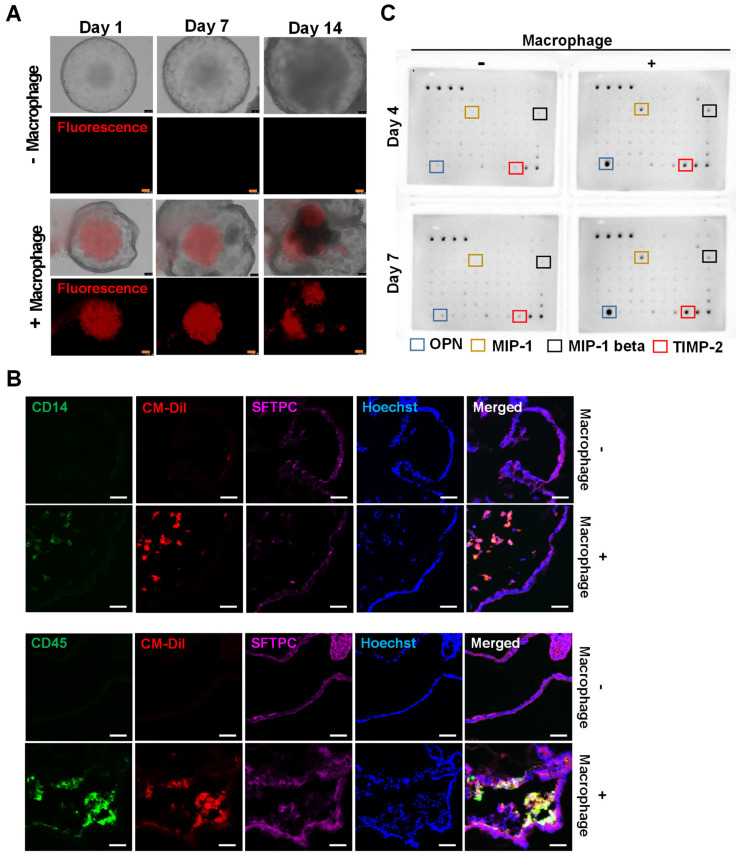
Induction of hESC-derived macrophages in lung organoid. (**A**) Representative phase-contrast and fluorescent dye images of lung organoids injected with macrophages at injection day 1 to 14. The red fluorescent cells indicated the injected macrophages. Scale bar = 100 μm. (**B**) Immunofluorescence images of lung organoids and injected macrophages. Scale bar = 50 μm. (**C**) Cytokine array images of lung organoid with or without macrophages at day 7 after injection.

**Figure 5 ijms-23-09211-f005:**
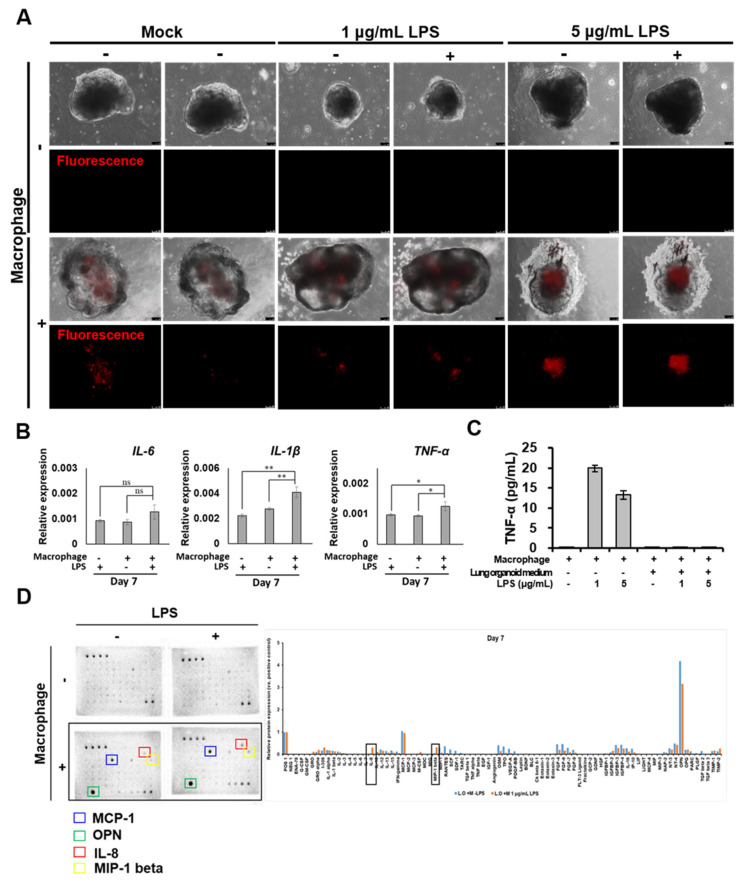
IL-8 and MIP-1 beta were increased in macrophage-included lung organoids under LPS stimulation. (**A**) Representative phase contrast and fluorescent dye images of macrophage-included lung organoid under LPS stimulation. Scale bar = 100 μm. (**B**) qRT-PCR analysis for IL-6, IL-1β, and TNF-α expression of lung organoid with or without macrophage and LPS stimulation at day 7 after injection. n = 3. * *p* < 0.05, ** *p* < 0.01 versus no treatment group. (**C**) The secretion rate of TNF-α in the hESC-derived iMAC cultured with macrophage cultured medium or, alveolar organoid medium stimulated with LPS. (**D**) Cytokine array images of lung organoid with or without macrophages and LPS stimulation at day 7 after injection.

## Data Availability

J.-H.K. gathered and maintained the data used in this investigation.

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
