# Peer review of "Human Pluripotent Stem Cell-Derived Alveolar Organoid with Macrophages"

_ijms, 2022, doi:10.3390/ijms23169211_

Round 1

Reviewer 1 Report

The manuscript can create interest because they point out the importance of immune cells in alveolar organoids for modelling inflammatory pulmonary diseases. The authors basically injected iMAC-like cells from pluripotent cells into differentiated alveolar organoids and cultured for more than 7 days and confirmed that inflammatory response can be induced. This has potential in generating lung disease models but there are some issues whose amendments will hopefully improve the manuscript.

Comments

The tone and choice of English words can be edited and rephrased to make it easy for the general reader to comprehend. the manuscript should be proofread by an English Editor.

·         Page 1 line 14 please read these articles and rephrase the sentence and a similar model was discussed here

Han, Y., Yang, L., Lacko, L.A. et al. Human organoid models to study SARS-CoV-2 infection. Nat Methods 19, 418–428 (2022). https://doi.org/10.1038/s41592-022-01453-y

Heo, HR., Hong, SH. Generation of macrophage containing alveolar organoids derived from human pluripotent stem cells for pulmonary fibrosis modeling and drug efficacy testing. Cell Biosci 11, 216 (2021). https://doi.org/10.1186/s13578-021-00721-2

·         page 2 line 48  reference 12 is a review article. please cite the original paper

Heo, HR., Hong, SH. Generation of macrophage containing alveolar organoids derived from human pluripotent stem cells for pulmonary fibrosis modeling and drug efficacy testing. Cell Biosci 11, 216 (2021). https://doi.org/10.1186/s13578-021-00721-2Heo, HR., Hong, SH. Generation of macrophage containing alveolar organoids derived from human pluripotent stem cells for pulmonary fibrosis modeling and drug efficacy testing. Cell Biosci 11, 216 (2021). https://doi.org/10.1186/s13578-021-00721-2

·         Page 2 line 66   Please rephrase the sentence “We expected 64 that this model system will provide a more precise understanding of human pulmonary 65 diseases.”

·        Page 2 line 73   Please rephrase “First, hPSCs were differentiated into 70 definitive endoderm cells. When the cells were filled in the dish, they were detached and 71 newly attached, and various growth factors were added to produce lung progenitor cells 72 (Figure 1B)

·       page 3 line 86  Please define the error bar. STD or SEM and include it in the legend

·         Please redo the statistics as the difference in most of the graphs does not appear significant.

·         Please provide much clearer images would be make the paper more easy to comprehend.

·         Please provide control plots (unstained and isotype will help here) for the facs plots.

·         Figure 5B It is relative to what?

·         Page 9 line 190  Please rephrase. it is difficult to understand “Glucocorticoids 188 (dexamethasone) inhibit several of the initial events in an inflammatory response and pro-189 mote the resolution of inflammation [25,26]”

·         Page 9 line 207  Please write a statement or two to conclude it.

The paper can be strengthened before publication.

Reviewer 2 Report

Ha-Rim Seo and colleagues performed original studies establishing a novel method of generation of alveolar organoids containing macrophages which are both derived from human induced pluripotent stem cells or human embryonic stem cells, and these macrophage-containing alveolar organoids were maintained for 14 days in which the macrophages possessed their inflammatory functions.  

Comments:

The title is required to be improved, for instance - Human pluripotent stem cell derived alveolar organoids with macrophages.

In conclusion is better to avoid abbreviations “iMACs in AO:” Please write iPSC-based macrophages in iPSC-based alveolar organoids.

Abbreviation iMACs is misleading, macrophages (i-MACs) (lines 45-46) “ is it a computer name?”

What does it mean i-MACs – it is not abbreviation, suggestion – iPSC-based macrophages (iMs).  

The same way for AO – it should be iPSC-based alveolar organoids (iOAs).

Should be - Mononuclear phagocytic cells (MPCs). Important to write the abbreviations in plurals (s).

It is better to use abbreviation hiPSCs or hESCs rather then PSCs in the text of the paper.

English is better to improve. In many parts, English is slangish.

Explanations for gene abbreviations are required: SOX17, SFPTB and SFPTC, NKX2.1, FoxJ1, PDPN, HSPC, TIMP-2, MIP-1β, TNF-α, MCP-1, VEGF and SCF and so on.

“4.2. AO differentiation” is correctly – iPSCs differentiation into alveolar organoids.

“4.3. iMAC differentiation” - iPSCs differentiation into macrophages.

In methods should be better explain what iPSCs and ESCs were used in sections 4.2. and 4.3.

Reviewer 3 Report

Authors: Ha-Rim Seo, Hyeong-Jun Han, Young-Woock Noh, Youngson Lee, Seung-Ju Cho, Jung-Hyun Kim 

Title: Human pluripotent stem cell derived alveolar organoid with macrophages derived from human pluripotent stem cell 

COMMENTS: 

In the submitted manuscript, the Authors describe creation and features of an organoid consisting of the alveolar epithelium and macrophages when both types of cells are derived from human pluripotent stem cells. Potential use of the created organoid is discuissed for modeling pathological states and drug effects in lungs. The manuscript is well written and illustrated; nevertheless, there are a couple of critical remarks: 

It is good that the Authors found in their model such typical response as production of IL-1β, MCP-1, and TNF-α after LPS treatment. However, what about phagocytosis? The function of macrophages as phagocytes should here be demonstrated and evaluated as well.   

I would recommend for the Authors to provide more detailed discussion for which situations their organoid may be helpful. Whether it can help to develop organoid-based lung implants? Whether it can be used for modeling lung fibrosis? Or cancer immunotherapy involving macrophages? 

Round 2

Reviewer 1 Report

They have satisfactorily answered all the questions raised.